# Integrating Meteorological Forcing from Ground Observations and MSWX Dataset for Streamflow Prediction under Multiple Parameterization Scenarios

Hamed Hafizi [1,2] and Ali Arda Sorman [1,*]

[1] Department of Civil Engineering, Eskisehir Technical University, Eskisehir 26555, Turkey
[2] Department of Hydraulics and Hydraulic Structures, Faculty of Water Resources and Environmental Engineering, Kabul Polytechnic University, Kabul 1010, Afghanistan
* Correspondence: asorman@eskisehir.edu.tr

**Abstract:** Precipitation and near-surface air temperatures are significant meteorological forcing for streamflow prediction where most basins are partially or fully data-scarce in many parts of the world. This study aims to evaluate the consistency of MSWXv100-based precipitation, temperatures, and estimated potential evapotranspiration (PET) by direct comparison with observed measurements and by utilizing an independent combination of MSWXv100 dataset and observed data for streamflow prediction under four distinct scenarios considering model parameter and output uncertainties. Initially, the model is calibrated/validated entirely based on observed data (Scenario 1), where for the second calibration/validation, the observed precipitation is replaced by MSWXv100 precipitation and the daily observed temperature and PET remained unchanged (Scenario 2). Furthermore, the model calibration/validation is done by considering observed precipitation and MSWXv100-based temperature and PET (Scenario 3), and finally, the model is calibrated/validated entirely based on the MSWXv100 dataset (Scenario 4). The Kling–Gupta Efficiency (KGE) and its components (correlation, ratio of bias, and variability ratio) are utilized for direct comparison, and the Hanssen–Kuiper (HK) skill score is employed to evaluate the detectability strength of MSWXv100 precipitation for different precipitation intensities. Moreover, the hydrologic utility of MSWXv100 dataset under four distinct scenarios is tested by exploiting a conceptual rainfall-runoff model under KGE and Nash–Sutcliffe Efficiency (NSE) metrics. The results indicate that each scenario depicts high streamflow reproducibility where, regardless of other meteorological forcing, utilizing observed precipitation (Scenario 1 and 3) as one of the model inputs, shows better model performance (KGE = 0.85) than MSWXv100-based precipitation, such as Scenario 2 and 4 (KGE = 0.78–0.80).

**Keywords:** meteorological forcing; MSWX; TUW; hydrologic modeling; mountainous basin; Turkey

## 1. Introduction

Precipitation and near-surface air temperature datasets with high spatial and temporal resolutions are crucial for many hydroclimatic studies such as hydrology, meteorology, agriculture, natural resources management, energy systems, hydrologic modeling, and risk assessment [1,2]. Moreover, air temperature variation plays a significant role in forming different precipitation patterns, which is essential for climate change studies and has high control on snowmelt and accumulation dynamics over snow-dominant regions, which causes seasonal streamflow regime oscillation [3–5]. Traditionally, air temperature is measured with a standard meteorological shelter placed 2 m above the ground [6] and the amount of precipitation is estimated by rain gauges located above the ground level [7,8]. However, denser meteorological stations are required to capture the spatio-temporal variability of precipitation and air temperature, presenting reliable weather observation over a region. Similarly, topographic complexity of highly elevated regions, cost of installation and maintenance, limitations in data-sharing policy, and potential hydro-political

tensions for transboundary river basins have been a great challenge for accurate weather observation in recent decades [9–12]. Furthermore, full or partial shortcomings of weather data observation often limited hydrological prediction in an ungauged basin (PUB) [13]. Therefore, an additional effort is needed to collect data if financial support is available or to explore alternative data sources that are more feasible. Hence, gridded weather datasets from numerous sources such as satellite and numerical weather prediction models' output are a kind of alternative to filling data scarcity over a particular region.

In recent years, a large number of gridded precipitation and air temperature datasets varying in spatial and temporal resolution have been provided for public use. Some of these datasets only present precipitation estimates, and the others are able to provide minimum, maximum, and average temperatures in addition to precipitation, where they are essential for potential evapotranspiration (PET) estimates and governing hydrologic models. Examples of gridded precipitation datasets are Precipitation Estimation from Remotely Sensed Information Using Artificial Neural Networks-Dynamic Infrared Rain Rate (PDIR-Now) [14], multi-source merging such as Multi-Source Weighted-Ensemble Precipitation (MSWEP) [15], and Climate Hazards group InfraRed Precipitation with Stations (CHIRPS) [16]. Some gridded datasets, such as Multi-Source Weather version 100 (MSWXv100) [17] and Climate Forecast System Reanalysis (CFSR) [18], are able to present both temperature and precipitation along with other climatological variables.

Regardless of how much physical information from the basin, including land use, land cover, soil, and topography, is incorporated or excluded by selecting a specific hydrologic model in the validation process, several studies have evaluated the accuracy of gridded weather datasets in driving hydrological models for streamflow prediction, which are mostly focused on precipitation rather than other meteorological forcing. For instance, Zhang et al. [19] evaluated three gridded precipitation datasets (CHIRPSv2.0, TMPA-3B42v7, and PERSIANN-CDR) over various basins having different climatic regimes in China using the Coupled Routing and Excess Storage (CREST) hydrologic model by considering two different scenarios: when the model parameters are calibrated by observed precipitation (Scenario-1) and each precipitation dataset individually (Scenario-2). Overall, precipitation datasets displayed high ability for daily streamflow simulation in Scenario-2, where TMPA-3B42v7 with a Nash–Sutcliffe Efficiency (NSE) of 0.96 and CHIRPSv2.0 with a NSE of 0.90 showed higher streamflow reproducibility over humid basins, while PERSIANN-CDR demonstrated the best performance (NSE = 0.67) in arid basins. Bhati et al. [20] evaluated TMPA-3B42v7 for streamflow prediction over the Upper Mahi basin using the Soil and Water Assessment Tool (SWAT) hydrologic model, considering four-year calibration/validation periods from 1998 to 2001 on a monthly scale. The result showed a strong correlation ($R^2 > 0.77$) of TMPA-3B42v7 with observed streamflow for runoff prediction. Gunathilake et al. [21] investigated the hydrologic utility of 12 precipitation datasets (TMPA-3B42v7, TMPA-3B42RT, PERSIANN, PERSIANN-CCS, PERSIANN-CDR, CHIRPSv2.0, CMORPH, IMERGHHFv06, MSWEPv1.1, APHRODITEv1801, APHRODITEv1901, and GPCCv1) over the Huai Bang Sai (HBS) basin in northeastern Thailand, utilizing the SWAT hydrologic model for the period of 2004–2014 considering the monthly time step. They demonstrated that all satellite-based precipitation datasets have a monthly NSE of more than 0.55, where MSWEPv1.1 and CHIRPSv2.0 were able to present streamflow close to observed discharge, especially for the calibration period (2004–2007), while APHRODITEv1901 (NSE > 0.53) showed higher reproducibility of streamflow compared to APHRODITEv1801, and GPCCv1 gauge-based gridded precipitation datasets. Satgé et al. [22] compared the hydrologic utility of 19 gridded precipitation datasets (CHIRPv2.0, ERA5, GSMaP-RTv6, IMERGHHEv06, IMERGHHLv06, MERRA2-FLX, TMPA-3B42RTv7, CMORPH-BLD, CMORPH-CRT, GSMaP-Adjv.6, IMERGHHFv.06, MERRA2-LND, PERSIANN-CSS-CDR, PERSIANN-CDR, TMPA-3B42v7, WFDEI-CRU, WFDEI-GPCC, CHIRPSv2, and MSWEPv2.2) over ten distinct basins of the Juruá watershed for ten years (2001–2010). They found that IMERGHHFv06 is the most reliable precipitation dataset for streamflow prediction with a median Kling–Gupta Efficiency (KGE) of 0.79 and

0.81 considering daily and monthly time steps. On the other hand, CHIRPv2.0 provided the least accuracy of streamflow simulation, with KGE values of 0.36 and 0.11 using GR4J and HyMOD, respectively.

In the same way, several authors evaluated various gridded precipitation datasets over different catchments in Turkey. For example, Karakoc and Patil [23] compared TMPA-3B42v7 and gauge-based precipitation estimates for monthly streamflow prediction in western Turkey's Kucuk Menderes river basin (3930 km$^2$). Both TMPA-3B42v7 and gauge-based precipitation estimates were used independently for EXP-HYDRO hydrologic model calibration (2003–2009) and validation (2010–2012). They found that gauge-based precipitation estimates showed higher performance (KGE = 0.82 for calibration; KGE = 0.76 for validation) compared to TMPA-3B42v7 (KGE = 0.54 for calibration; KGE = $-1.08$ for validation) and the post bias correction of TMPA-3B42v7 with ground gauge data considerably increased TMPA-3B42v7 dataset performance (KGE = 0.81 for calibration; KGE = 0.62 for validation) for streamflow prediction. Similarly, Uysal and Şorman [24] employed the Multilayer Perceptron (MLP) Artificial Neural Network (ANN) model to evaluate PERSIANN and PERSIANN-CDR for streamflow prediction over the upper Euphrates river basin (Karasu) in Turkey. The results indicated that both precipitation datasets exhibited high streamflow reproducibility when post bias correction was applied to the mentioned precipitation datasets, and the NSE increased from 0.38 to 0.68 and 0.48 to 0.61 for PERSIANN-CDR and PERSIANN for the validation period (2009–2011). Finally, Hafizi and Sorman [25] evaluated the hydrological utility of 13 gridded precipitation datasets (CPCv1, MSWEPv2.8, ERA5, CHIRPSv2.0, CHIRPv2.0, IMERGHHFv06, IMERGHHEv06, IMERGHHLv06, TMPA-3B42v7, TMPA-3B42RTv7, PERSIANN-CDR, PERSIANN-CCS, and PERSIANN) utilizing the TUW (Technical University of Wien) hydrologic model in a mountainous basin for five water years (2015–2019) at a daily time step. The results indicated that CPCv1, MSWEPv2.8, CHIRPSv2.0, and CHIRPv2.0 datasets outperformed the others when model parameters are set for the ground observations.

In addition, several authors evaluated the spatio-temporal consistency of gridded precipitation [26–33] and temperature [34–37] datasets over various regions by direct comparison with observed data. The results infer that temperature datasets are much more stable, showing higher performance compared to precipitation datasets over time and space for the daily and monthly time steps.

Consequently, it appears from the literature that many gridded precipitation and temperature datasets are evaluated either directly by observed precipitation and temperature data or primarily focused on only utilizing precipitation datasets for streamflow prediction, where some basins show a high record of precipitation estimates. At the same time, other meteorological forcings may either be scarce or even non-existent. Hence, we see the necessity of developing a new approach that exploits not only precipitation dataset but also other meteorological forcing such as temperatures and estimated evaporation data from a single non-ground source by integrating it with the existing ground observations for streamflow simulation.

This study aims to evaluate the consistency of MSWXv100-based precipitation, temperatures, and estimated potential evapotranspiration (PET) by direct comparison with observed data and by integrating meteorological forcing from MSWXv100 datasets along with ground observations for streamflow reproducibility under distinct model parameterization scenarios.

The structure of this paper is organized as follows: Section 1 presents a detailed introduction to weather datasets and their hydrological utility. Section 2 provides information on materials and methods. Section 3 shows results and comprehensive discussions, and conclusions are presented in Section 4.

## 2. Materials and Methods

### 2.1. Study Area

This study was carried out in the Karasu River basin (Figure 1), a mountainous catchment in Turkey located between 38°58′ E and 41°39′ E and 39°23′ N to 40°25′ N. Moreover, the study area is in the headwaters of the largest basin (Euphrates) in the eastern part of the country, and its elevation varies from 1130 to 3500 m. Karasu basin has a drainage area of around 10,250 km$^2$, and the runoff volume is continuously measured by stream gauging station (E21A019) at the outlet. Annual average runoff recorded at the outlet of the basin is around 230.8 mm, where annual total precipitation is estimated as 431 mm and mean temperature recorded is around 9.2 °C in the basin. The basin receives precipitation in the form of rain and snow; once the temperatures rise during the spring and early summer seasons, snowmelt contributes up to two-thirds of the annual runoff volume. Since the cascade dams are located downstream of the basin, accurate estimation of runoff at the outlet of Karasu basin resulting from snowmelt is particularly vital for flood forecasting, reservoir management, hydropower generation, irrigation, and water supply [24]. Furthermore, Karasu catchment is one of the pilot basins for nationally and internationally funded scientific projects [38–43]. Hence, it is crucial to understand the flow regime changes and accurate estimation of runoff volume and its temporal distribution over the region.

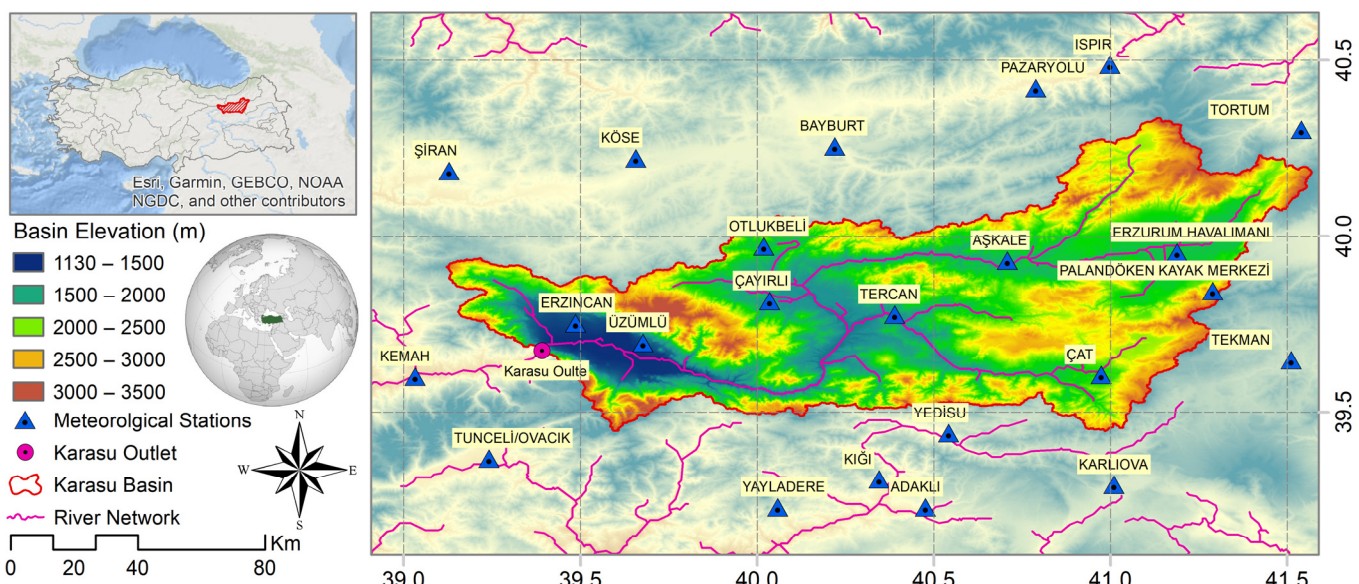

**Figure 1.** Geographical location of the study area (Karasu basin), basin elevation (m) ranges, and distribution of hydrometeorological stations.

### 2.2. Data

In this study, three types of hydrometeorological datasets were utilized; observed precipitation and temperature datasets (benchmark/reference), precipitation and temperatures from the MSWXv100 dataset, and observed streamflow required for hydrological modeling. The observed daily precipitation and temperatures (minimum, maximum, and average) from 23 meteorological stations were provided by the State Meteorological Directorate (MGM), and streamflow data for Kemah Boğazı (E21A019) stream gauging station at the outlet of basin were obtained from State Hydraulics Works (DSI) of Turkey.

The Multi-source Weather (MSWX) version 100 is an operational global near-surface gridded meteorological dataset with a high spatial (0.1°) and temporal (3-hourly) resolution and a short lag time (3 h of latency). Moreover, MSWXv100 provides medium-range (up to ten days) and long-range (up to seven months) bias-corrected forecast ensembles and takes advantage of other datasets such as CHELSA, CRU TS, ERA5, FLUXNET, GDAS, GEFS,

GHCN-D, GSOD, and ISD. The dataset presents ten meteorological variables (precipitation, average, minimum and maximum air temperature, surface pressure, relative and specific humidity, wind speed, and downward shortwave and longwave radiation) and its historical records are available from January 1979 onward. MSWXv100 is utilized in a wide range of applications, including hydrological modeling, water resources management, flood forecasting, drought monitoring, and disease tracking at the global/regional scale [17]. MSWXv100 data can be found at: www.gloh2o.org/mswx (accessed on 10 July 2022).

This study was conducted for five water years (October 2014 to September 2019) based on the availability of selected datasets.

*2.3. TUW Hydrologic Model and Streamflow Simulation Scenarios*

In this context, the TUW conceptual hydrologic model developed by the Technical University of Vienna is selected for streamflow simulation. The TUW model was built on a similar structure to the HBV (Hydrologiska Byråns Vattenbalansavdelning) model and tested over 320 Austrian catchments [44,45]. The model has been used in several studies [44,46–50], and its related equations can be found in Parajka et al. (2007). The TUW model has 15 parameters (Table 1) operating at a daily time step and is able to simulate snow, soil moisture, and runoff processes. The model inputs are total daily precipitation (mm), mean daily near-surface temperature (°C), and daily potential evapotranspiration (mm). The temperature-based Hargreaves–Samani method is utilized to calculate potential evapotranspiration, which uses the daily minimum and maximum near-surface air temperature [51]. Moreover, the model parameters are calibrated by hydroPSO global optimization algorithm [52,53] that implements the latest version of the Particle Swarm Optimization (PSO) technique which has been successfully used to calibrate numerous hydrological and environmental models [52,54–56].

**Table 1.** TUW model parameter properties. Abbreviations in the process column represent: S, snow; SM, soil moisture; R, runoff.

| ID | Description | Units | Process | Range |
|----|-------------|-------|---------|-------|
| SCF | Snow correction factor | - | S | 0.9–1.5 |
| DDF | Degree-day factor | mm/°C /day | S | 0.0–5.0 |
| Tr | Temperature threshold above which precipitation is rain | °C | S | 1.0–3.0 |
| Ts | Temperature threshold below which precipitation is snow | °C | S | −3.0–1.0 |
| Tm | Temperature threshold above which melt starts | °C | S | −2.0–2.0 |
| LPrat | Parameter related to the limit for PET | - | SM | 0.0–1.0 |
| FC | Field capacity | mm | SM | 0.0–600 |
| BETA | Non-linear parameter for runoff production | - | SM | 0.0–20 |
| cperc | Constant percolation rate | mm/day | R | 0.0–8.0 |
| k0 | Storage coefficient for very fast response | day | R | 0.0–2.0 |
| k1 | Storage coefficient for fast response | day | R | 2.0–30 |
| k2 | Storage coefficient for slow response | day | R | 30–250 |
| lsuz | Threshold storage state | mm | R | 1.0–100 |
| bmax | Maximum base at low flows | day | R | 0.0–30 |
| croute | Free scaling parameter | $day^2/mm$ | R | 0.0–50 |

The dataset is divided into calibration (October 2014 to September 2016) and validation (October 2016 to September 2019) periods. It is worth mentioning that just like in many parts of the world, several mountainous basins in Turkey have very limited or sometimes non-existent meteorological forcing where only streamflow records are available. Generally, two scenarios have been utilized for streamflow simulation using Gridded Precipitation Datasets (GPDs) in the literature. In Scenario 1, the model is first calibrated by observed meteorological data, then only observed precipitation is replaced by each GPD, and in Scenario 2, the model is directly calibrated by each GPD [19,57–60]. However, in Scenario 1, the model parameters are calibrated by observed meteorological data, which may not be the optimal parameter set for GPDs. In Scenario 2, the model parameters are calibrated by each

GPD individually, where other meteorological forcing such as observed temperature and PET are still a part of the model input. Moreover, there are many basins with scarce/non-existent meteorological data, and only streamflow values exist, which makes it harder to apply the above two scenarios. The novelty of this study is to simulate streamflow by considering four independent scenarios. Initially, it is assumed that fully observed meteorological data exists, but later on, it is considered such that one or more observed meteorological forcings do not exist, and we attempted to fill this gap with the MSWXv100 dataset. Hence, the following four scenarios can be illustrated as follows. In Scenario 1, the model parameters are calibrated/validated entirely based on observed data (observed precipitation, observed mean temperature, and estimated PET from observed data). In Scenario 2, the model parameters are calibrated/validated by MSWXv100 precipitation, observed mean temperature, and calculated PET based on observed data. In Scenario 3, the model parameters are calibrated/validated by observed precipitation and MSWXv100 mean temperature, and the calculated PET is based on MSWXv100 data. Furthermore, in Scenario 4, the model parameters were calibrated entirely based on MSWXv100 meteorological data (MSWXv100 precipitation, MSWXv100 mean temperature, and estimated PET from MSWXv100 data).

### 2.4. Evaluation Approach

The modified Kling–Gupta Efficiency (KGE) (Equation (1)), including its three components, namely Pearson correlation coefficient (r), the ratio of bias (β), and variability ratio (γ), were utilized [61,62].

$$\text{KGE} = 1 - [(r-1)^2 + (\beta-1)^2 + (\gamma-1)^2]^{0.5} \tag{1}$$

Among them, r is the Pearson correlation coefficient, which represents the linear correlation between observed and climate dataset (Equation (2)); β measures the amount of overestimation (β > 1) or underestimation (β < 1) of selected dataset (Equation (3)); and γ is a relative measure of dispersion (Equation (4)). Moreover, μ and δ are the distribution mean and standard deviation where s and o indicate estimated and observed values.

$$r = \frac{1}{n}\sum_{1}^{n}(o_n - \mu_0)(s_n - \mu_s)/(\delta_o \times \delta_s) \tag{2}$$

$$\beta = \frac{\mu_s}{\mu_o} \tag{3}$$

$$\gamma = (\delta_s \times \mu_o)/(\mu_s \times \delta_o) \tag{4}$$

KGE is relatively a new objective function providing an overall accuracy of the selected dataset by balancing the contributions of correlation (r), ratio of bias (β), and variability ratio (γ). This group of metrics has been used for many studies related to climate dataset validation in recent year [26,29,40,63]. In the same way, the Hanssen–Kuiper (HK) skill score (Equation (5)) is employed to assess the detectability strength of MSWXv100 for different precipitation intensities and it has been used in previous studies related to GPDs detectability assessment [9,40]. In Equation (5), M (Miss) shows that the observed precipitation is not detected, F (False) represents the condition when a precipitation is detected but not observed, H (Hit) indicates the correctly detected observed precipitation, and CN (Correct Negative) shows that a no precipitation event is detected.

$$\text{HK} = \frac{(H \times CN) - (F \times M)}{(H+M)\,(F+CN)} \tag{5}$$

The detectability strength of MSWXv100 precipitation is evaluated based on daily precipitation from observed and MSWXv100, which is classified into the five following thresholds: no/tiny precipitation (less than 1 mm/day), light precipitation (1–5 mm/day),

moderate precipitation (5–20 mm/day), heavy precipitation (20–40 mm/day), and violent precipitation (more than 40 mm/day) [64,65].

In addition, the Nash–Sutcliffe Efficiency (NSE) [66] along with the KGE are employed to quantify the accuracy of simulated streamflow obtained from four distinct scenarios by combining observed and MSWXv100 datasets (Equation (6)).

$$\text{NSE} = 1 - \frac{\sum_{i=1}^{n} \left( Q_i^s - Q_i^o \right)^2}{\sum_{i=1}^{n} \left( Q_i^o - \overline{Q_i^o} \right)^2} \tag{6}$$

where n is the sample size of the observed or calculated streamflow. $Q_i^o$ and $Q_i^s$ present the observed and simulated streamflow, and $\overline{Q_i^o}$ presents the mean observed streamflow. All selected metrics have their optimum value at unity.

## 3. Result and Discussion

### 3.1. Evaluation of Meteorological Data at the Regional Scale

Figure 2 presents a comprehensive statistical analysis of meteorological data at the regional scale derived from observed and MSWX datasets. Figure 2a displays daily temperature distributions in the form of box and whisker plots. Considering the observed temperatures, the region experiences a 9.0 °C daily mean average temperature while this amount decreases to 3.3 °C for daily mean minimum temperature and increases to 16.1 °C for daily mean maximum temperature. Moreover, the observed daily maximum temperature shows a wider distribution compared to the daily minimum and average temperature records.

Furthermore, MSWX was able to exhibit close daily average temperature measurements to the observed and no significant differences were detected between their medians. However, it shows slightly higher daily mean minimum (4.55 °C) and lower daily mean maximum (15.7 °C) temperatures, comparatively. Referring to Figure 1, most of the 23 stations in the region are located at low elevated, flat/near to flat, and open areas (plateau regions). Hence, this condition may have less effect on average temperatures but more on the extremes (minimum and maximum). In flat and open areas, minimum temperatures may be measured cooler and at the same time maximum temperatures warmer than usual which may be the cause–effect relationship for the larger deviations at the extremes. Figure 2b displays the mean daily precipitation, estimated PET, and average temperature obtained from observed and MSWX datasets over five hydrological years (2015–2019). Overall, MSWX shows higher precipitation and lower PET compared to observed data in each distinct water year. Moreover, the region carried the lowest daily precipitation amount in 2017. In the same way, the daily mean average temperature from observed and MSWX shows the same pattern for the selected five years with a slight overestimate by MSWX. The region received a lower average temperature during 2017 and a higher one in 2018. Furthermore, the scatter plot (Figure 2c) indicates that MSWX daily precipitation has a large variation compared to the observed which can also be confirmed by the low coefficient of determination ($R^2 = 0.35$). Finally, MSWX precipitation shows a larger residual distribution with a slight overestimation for each water year, while estimated PET from MSWX presents a lower variability residual and a small underestimation of PET over selected years (Figure 2d).

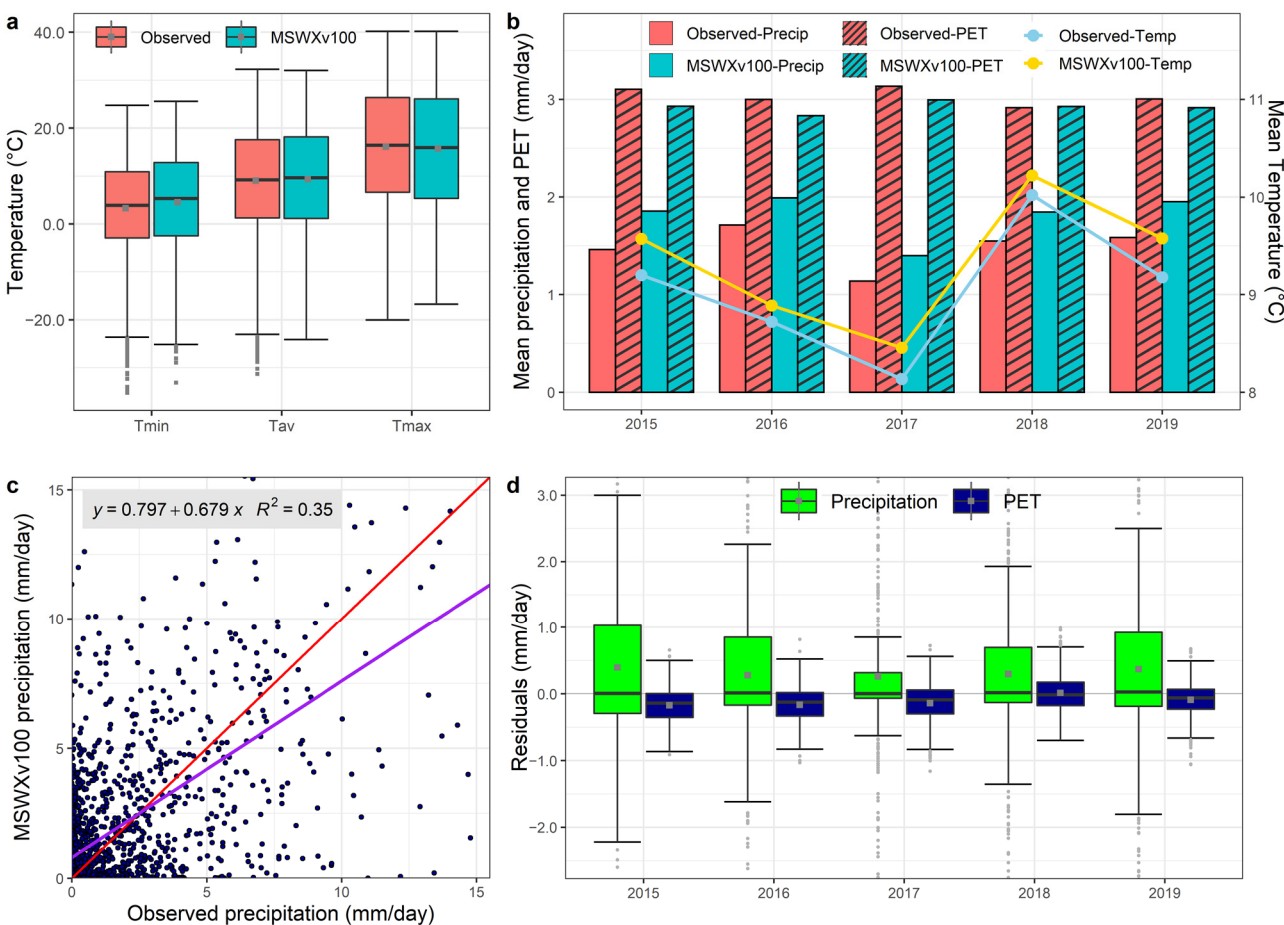

**Figure 2.** Comparison of daily meteorological data from MSWXv100 against observed data. (**a**) Temperatures (min, average, max) at the regional scale, (**b**) mean daily precipitation, PET, and temperature over hydrologic years, (**c**) daily precipitation scatter plot, and (**d**) daily residuals for MSWXv100 and PET over five water year (2015–2019).

### 3.2. Reliability and Detectability Strength of MSWXv100 Dataset at Daily Time Step

At the regional scale, considering other meteorological forcing such as temperatures and PET, the ability of MSWX precipitation to quantify the daily precipitation amount is relatively lower, having a median KGE value of 0.53 (Figure 3a). Furthermore, MSWX precipitation has a lower correlation (median of 0.59), a higher bias ratio (median of 1.21), and a slightly lower variability ratio (median of 0.95). However, comparing the MSWX precipitation performance in this study with satellite-based, reanalysis, and multi-source merging gridded precipitation datasets evaluation over Karasu river basin with the same station distribution and time window [25], MSWX precipitation is the only dataset which shows the highest performance (median KGE of 0.53) over the selected catchment. Overall, meteorological forcing, such as temperatures and estimated PET from the MSWX dataset, displays much higher performance (median of KGE > 0.90) and is close to optimum value (unity) for each indicator. In addition, the daily average temperature from MSWX shows the highest performance with 0.97 KGE among selected meteorological variables. In consideration of five precipitation intensity thresholds (Figure 3b), MSWX precipitation shows lower frequency for daily precipitation events less than 1 mm compared to observed precipitation, while it shows higher precipitation frequency for light (1–5 mm/day) and moderate (5–20 mm/day) precipitation, and close frequency to observed heavy (20–40 mm/day) and violent (more than 40 mm/day) storms. Moreover, MSWX shows the highest detectability strength for daily precipitation of less than 1 mm and its detectability decreases by increasing precipitation intensity, which is generally the case in literature [25,26,31,65].

Usually, detecting an event with a low occurrence probability is harder than identifying an event with a high occurrence. On the other hand, 5 diverse intensity thresholds instead of a binary precipitation/no-precipitation event can be considered as a demanding classification scheme. This, along with larger frequency differences between light/moderate intensities (Figure 3), could be attributed as the main reasons why MSWX demonstrates higher detectability for moderate precipitation compared to light precipitation. Such results are also observed in literature for other gridded precipitation products [25,65,67].

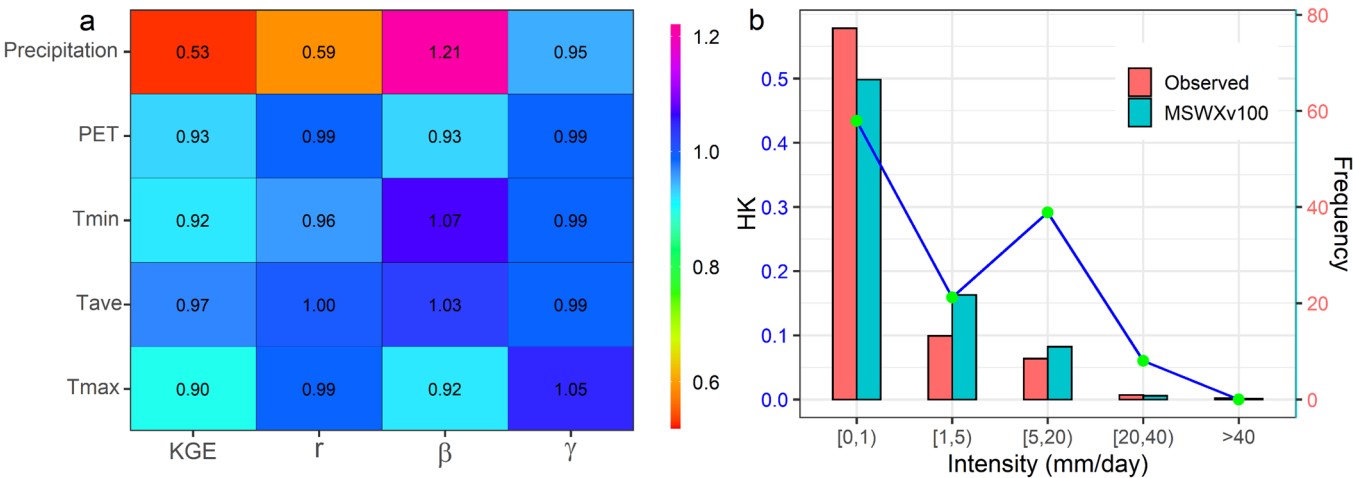

**Figure 3.** Reliability of MSWXv100 dataset at the regional scale. (**a**) Performance of MSWXv100 daily precipitation, temperatures, and estimated PET expressed in the form of KGE and its components, and (**b**) observed and MSWXv100 precipitation frequency and MSWXv100 precipitation detectability strength expressed in the form of Hanssen–Kuiper (HK) skill score.

### 3.3. Streamflow Simulation under Individual Model Parameterization Scenarios

In this context, the TUW model was utilized to predict streamflow at the outlet of Karasu basin using observed and MSWX data as meteorological forcing under four independent model parameterization scenarios considering five hydrological years (2015–2019). In each scenario, the model is first calibrated for two years (2015–2016), then validated by obtaining behavioral parameter sets and optimum model parameters for another three years (2017–2019). In the same way, the model was verified by considering the entire period (2015–2019).

### 3.3.1. Calibration of Model Parameters and Uncertainty Analysis

The model parameters for four scenarios are calibrated by the hydroPSO package in the R programming environment, which includes the Particle Swarm Optimization (PSO) algorithm [52,53]. The PSO is a population-based stochastic optimization approach utilized to explore a delimited search space with a swarm of particles to find the best set of parameters required for the maximization of user-defined objective function. Here we selected the Kling–Gupta objective function to be maximized. Considering the number of parameters (15), the number of particles in the swarm is selected as 80 and the maximum number of iterations is set to 50. Hence, 4000 model runs are obtained for each scenario. Furthermore, in this study, we selected only those behavioral parameter sets whose goodness of fit is greater than 0.3 (KGE > 0.3). According to the user-defined behavioral threshold (beh.th > 0.3), 2572 parameter sets were utilized to map the variation of each parameter in the form of box and whisker plots for four scenarios.

Figure 4 shows the density distribution of user-defined behavioral parameter sets along with optimum parameter values within each parameter upper and lower bound (Table 1) for four scenarios. Regardless of the whiskers and selected scenarios, when the selected algorithm (PSO) is forced to search for behavioral parameter sets within the range of each parameter value, sometimes the behavioral parameter sets vary in a smaller domain

(interquartile) than the upper and lower threshold of a specific parameter, while some behavioral parameter sets vary in a wider range (interquartile) close to the parameter range. For example, the user-defined behavioral parameters set for the field capacity (FC) parameter vary between 40 mm and 210 mm in four scenarios where the range of FC changes from 0 mm to 600 mm. In the same way, the storage coefficient for slow response (K2) range is from 30 days to 250 days, where the interquartile for behavioral parameter sets considering all scenarios are obtained in a certain interval (30–100 day). On the other hand, some parameters differed in their behavior under different parameterization scenarios. For example, the user-defined behavioral parameter sets for the non-linear parameter for runoff production (Beta) in Scenario 2 ($P_{MSWX}$, $T_{Obs}$, and $PET_{Obs}$) and Scenario 4 ($P_{MSWX}$, $T_{MSWX}$, and $PET_{MSWX}$) show larger fluctuations (interquartile) compared to Scenario 1 ($P_{Obs}$, $T_{Obs}$, and $PET_{Obs}$) and Scenario 3 ($P_{Obs}$, $T_{MSWX}$, and $PET_{MSWX}$). Moreover, some of the model parameters demonstrate close behavioral parameter sets (interquartile) for a different parameter value. For instance, the interquartile range of the Snow Correction Factor (SCF) parameter for Scenario 1, Scenario 2, and Scenario 3 shows almost the same width in the higher SCF values, whereas Scenario 4 shows the same interquartile of behavioral parameter sets in lower SCF values, comparatively. Hence, the interquartile for behavioral parameter sets provides the opportunity for the modification of some model parameter ranges, over Karasu basin that needs an explicit endeavor, which is beyond the scope of this study.

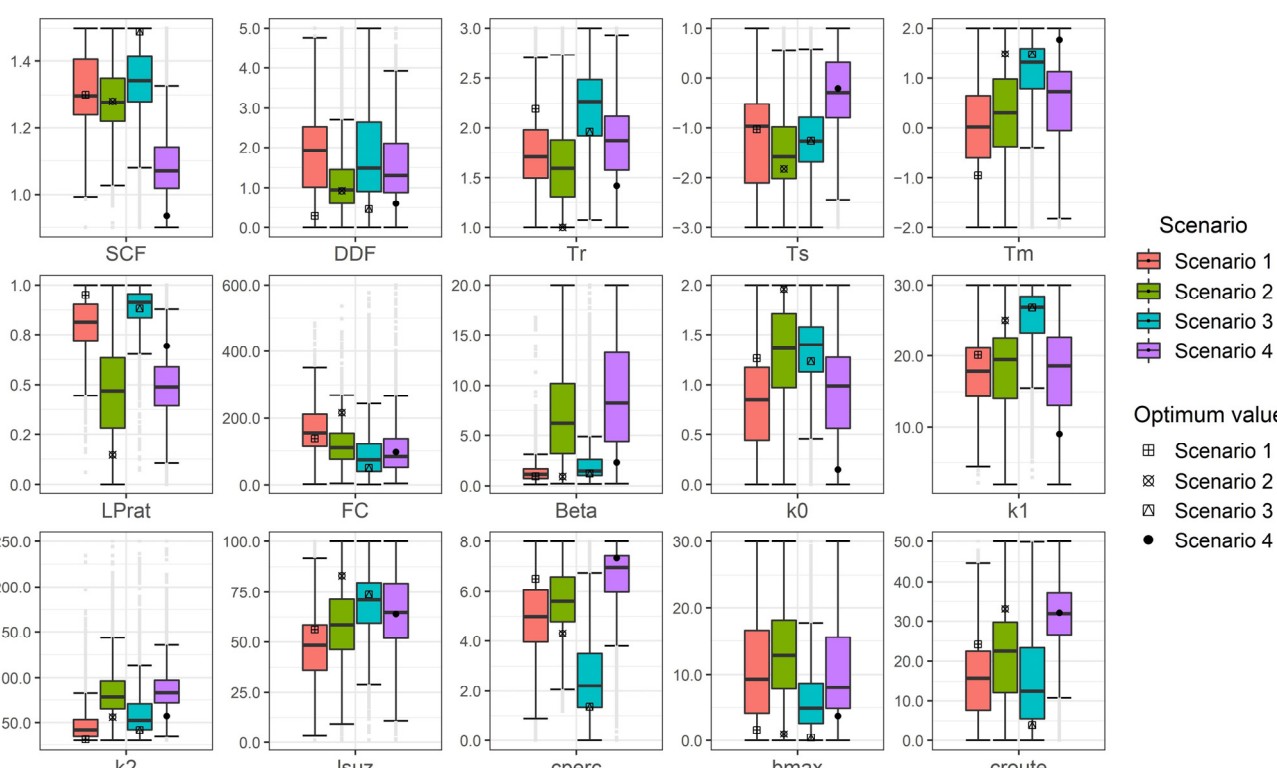

**Figure 4.** Box-and-whisker plots of behavioral parameter sets (KGE > 0.3) versus parameter values (2572 sets). The points indicate the optimum parameter value found through the optimization where the lower and upper edges of the box represent the 25th and 75th percentile values, respectively, while the "whiskers" represent the minimum and maximum values.

Additionally, based on the user-defined behavioral parameter sets (2572) obtained during the calibration period by setting the goodness of fit (KGE) greater than 0.3, the weighted quantile for model parameters are computed to provide an estimate of the uncertainty in each model parameter. The 95 percent prediction uncertainty (95PPU) at the 2.5% and 97.5% levels of the cumulative distribution for each model parameter are

obtained by multiplying the standard quantile derived for a certain parameter based on all user-defined behavioral parameter sets by the corresponding goodness of fit (KGE) values. Figure 5 shows the 95PPU, median, and best values of model parameters. Considering the lower (2.5%) and upper (97.5%) bound values of 95PPU obtained from the cumulative distribution of model parameters for four scenarios against parameter values, the model parameters demonstrate a distinct uncertainty range for each scenario. For example, in Scenarios 1 and 3, the uncertainty bound for some parameters (e.g., Beta, K1, LPrat) varies in a small range, or in other words, based on model parameter thresholds, the distance between the lower and upper bound of 95 PPU is smaller than for the same parameters in Scenarios 2 and 4. This can be attributed to the fact that if observed precipitation is used as input, then the uncertainty bound for the model parameters decreases and there seems to be a relatively larger uncertainty when the MSWX precipitation is utilized as one of the meteorological forcing (Scenario 2 and Scenario 4).

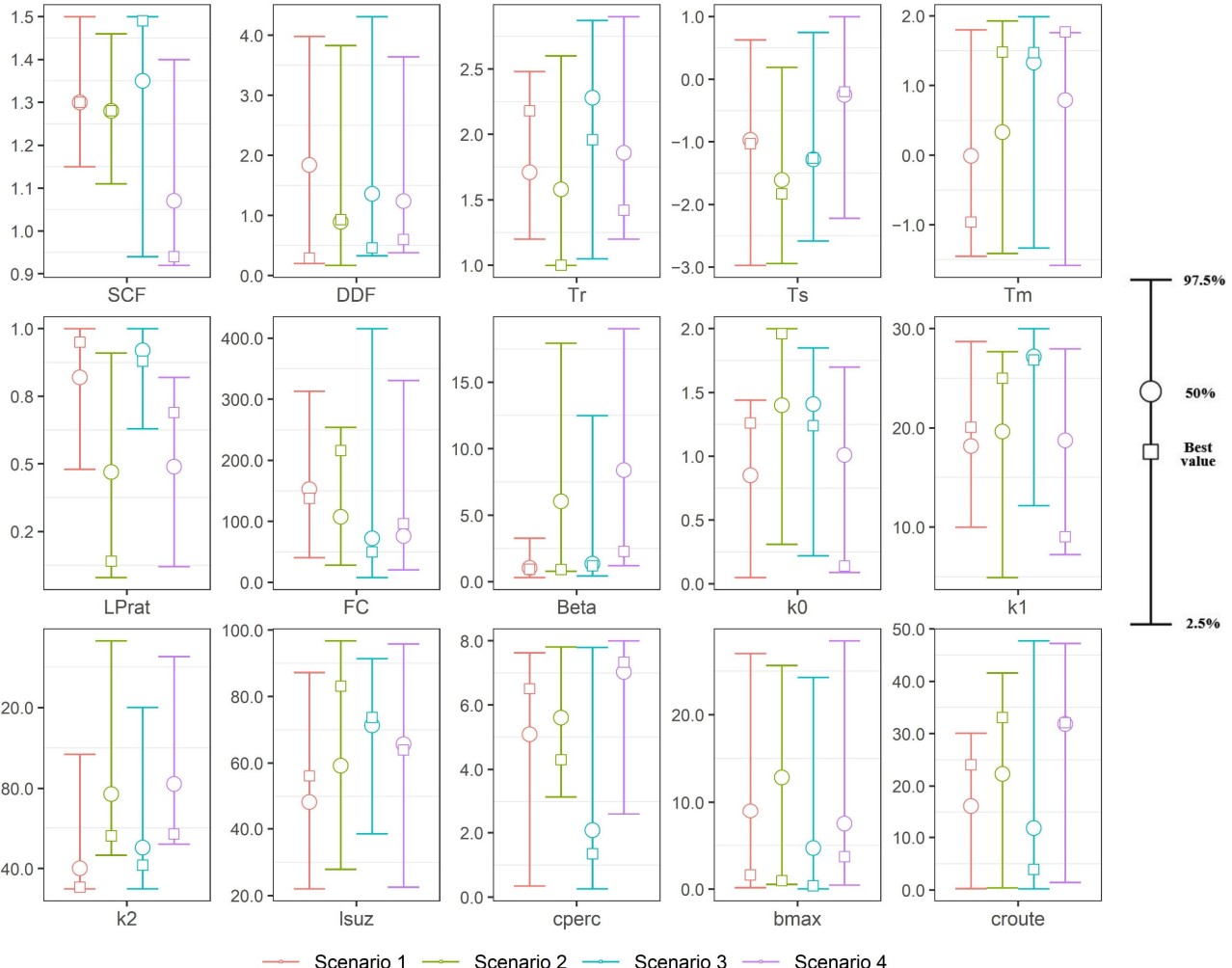

**Figure 5.** The 95 percent prediction uncertainty (95PPU) for model parameters during the calibration period (2015–2016) obtained by 2572 user-defined behavioral parameter sets (KGE > 0.3).

### 3.3.2. Streamflow Simulation with 95PPU

The P-factor and R-factor are statistical indicators which evaluate the degree of uncertainty in streamflow simulation. In general, the P-factor is calculated as the percentage of measured data captured by 95% prediction uncertainty (95PPU), which is estimated at the 2.5% and 97.5% levels of cumulative distribution of an output variable (in this case, simulated streamflow), where the R-factor shows the average thickness of 95% prediction

uncertainty (95 PPU) in the output variable. In general, a P-factor of 1 and an R-factor of zero is a simulation which exactly corresponds to observed streamflow data.

The PSO algorithm was utilized to estimate the 95 percent prediction uncertainty (95PPU) at the 2.5% and 97.5% levels of the cumulative distribution of streamflow simulations obtained from 2572 behavioral parameter sets.

Table 2 shows the results of model output uncertainty (95PPU) considering four scenarios for calibration, validation, and entire periods and Figure 6 displays the observed streamflow, simulated streamflow based on optimum model parameters (see Figures 4 and 5), and 95 percent prediction uncertainty (95PPU) bound for the calibration and validation periods.

**Table 2.** The 95PPU of streamflow simulation for four scenarios expressed in the form P factor and R factor.

| P/R-Factors | Scenario 1 | Scenario 2 | Scenario 3 | Scenario 4 | Temporal | Time Window |
|---|---|---|---|---|---|---|
| P-factor | 0.84 | 0.74 | 0.84 | 0.78 | Calibration Period | 2015–2016 |
| R-factor | 1.24 | 1.28 | 1.30 | 1.27 | Calibration Period | 2015–2016 |
| P-factor | 0.66 | 0.58 | 0.66 | 0.60 | Validation Period | 2017–2019 |
| R-factor | 0.95 | 1.06 | 0.99 | 1.01 | Validation Period | 2017–2019 |
| P-factor | 0.73 | 0.65 | 0.74 | 0.67 | Entire period | 2015–2019 |
| R-factor | 1.05 | 1.13 | 1.09 | 1.09 | Entire period | 2015–2019 |

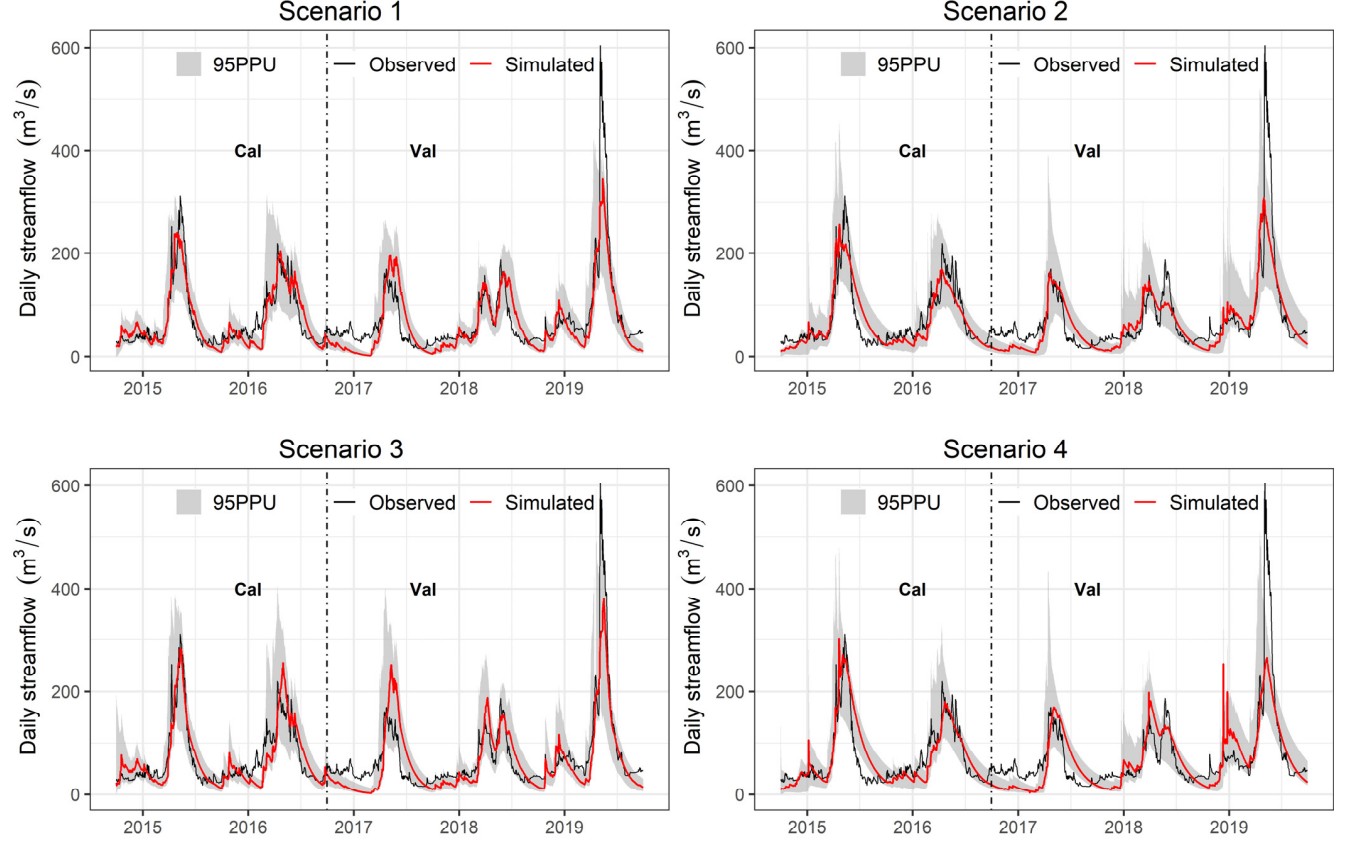

**Figure 6.** Observed and simulated discharge based on optimum model parameter value and the uncertainty bound (95PPU) considering four scenarios for the calibration (2015–2016) and validation (2017–2019) periods.

Considering the entire period (2015–2019), 73% of streamflow observations with a relative width (*R* factor) of 1.05 are enveloped by 95PPU in Scenario 1 ($P_{Obs}$, $T_{Obs}$, and $PET_{Obs}$), while in Scenario 2 ($P_{MSWX}$, $T_{Obs}$, and $PET_{Obs}$), only 65% of streamflow observations with a relative width of 1.13 are covered by 95PPU. In the same way, 74% of

streamflow observations with a relative width of 1.09 fall within the 95PPU considering Scenario 3 ($P_{Obs}$, $T_{MSWX}$, and $P_{MSWX}$) and only 67% of streamflow observations are covered by the 95PPU bound with a relative width of 1.09 for Scenario 4 ($P_{MSWX}$, $T_{MSWX}$, and $PET_{MSWX}$).

From the results, two important conditions can be spotted: precipitation data have a significant effect on the uncertainty of simulated streamflow compared to average temperature and PET meteorological forcing. For example, using observed precipitation, temperature, PET (Scenario 1), and MSWX-based temperature and PET (Scenario 3) resulted in covering a high portion of the observed streamflow by 95PPU bound. In the same way, considering the uncertainty performance indicators (Table 2), all scenarios show high P factors and a larger relative width (R-factor) for the calibration period compared to the validation period. This can be attributed to the fact that the behavioral parameter sets were obtained for the calibration period and the same sets were used for model verification.

Figure 7 presents the scatter plot of simulated streamflow against observed streamflow for four selected scenarios considering the entire period (2015–2019) time window. The color ramp demonstrates simulated streamflow bias for each scenario. For the low flows, the 2017 hydrologic year seems to depart the most from the observed measurements and for the high flows, the 2019 year shows the highest inaccuracy (Figure 6). Simulated streamflow in four scenarios mostly underestimate observed flow when discharge is more than 300 $m^3$/s and the highest underestimation is obtained when the model is calibrated entirely by MSWX data (Scenario 4). Overall, however, when observed precipitation is used as one of the meteorological forcing (Scenario 1, 3), the model shows higher streamflow reproducibility. That is why the coefficient of determination ($R^2$) decreases from 0.77 to 0.67 as more meteorological forcing is being utilized by MSWX dataset within the four scenarios.

Figure 8 shows how well the observed and MSWX datasets simulate streamflow in a certain scenario for the calibration (2015–2016), validation (2017–2019), and entire study (2015–2019) time periods. Overall, when observed data are utilized for streamflow simulation (Scenario 1), it comes with the best NSE, KGE, and correlation with observed streamflow for calibration/validation and entire study period. Moreover, Scenario 3 ($P_{Obs}$, $T_{MSWX}$, $PET_{MSWX}$) shows the second-best performance for streamflow simulation, followed by Scenario 2 ($P_{MSWX}$, $T_{Obs}$, $PET_{Obs}$). When the model is run entirely based on MSWX datasets, the simulated streamflow still shows high accuracy although coming last among all the scenarios for each period. Overall, the existence of bias in MSWX precipitation (Section 3.2 and Figure 3) has a high influence on streamflow simulation compared to other meteorological forcing. Using observed precipitation either with observed temperature and PET or MSWX temperature and PET show high streamflow reproducibility. On the other hand, model performance slightly degrades if partly or complete MSWX datasets are employed although still within acceptable limits. Moreover, looking at the direct comparison of the MSWX precipitation dataset with observed precipitation (Figures 2c and 3a), MSWX precipitation shows higher performance for streamflow simulation than its performance obtained by direct comparison with ground data. This can be attributed to the fact that precipitation is the most uncertain part of the hydrologic cycle compared to streamflow while its distribution varies over time and space faster than the variation of streamflow measured at a single point (outlet). Moreover, for streamflow simulations, precipitation is probably the most critical meteorological forcing for hydrological models. However, other model forcings such as temperature and potential evapotranspiration, which show promising performance (Figure 3) in direct comparison with ground observations, may also have an influence for this study considering a significant snowmelt process in a mountainous basin.

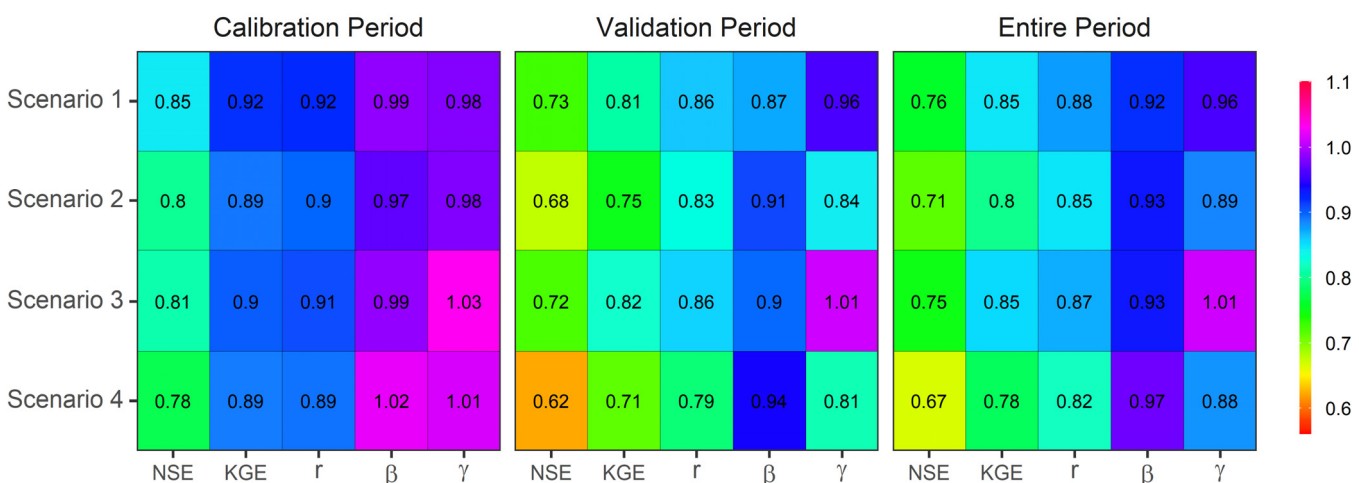

**Figure 7.** Scatter plot of simulated versus observed streamflow for four scenarios considering the entire period (2015–2019). The color ramp in each scatter plot shows the presence and magnitude of bias between simulated and observed streamflow.

**Figure 8.** Performance of observed and MSWXv100 datasets in generating daily streamflow simulation for the calibration (2015–2016), validation (2017–2019), and entire (2015–2019) periods under four model calibration scenarios.

## 4. Summary and Conclusions

Based on benchmark (observed) meteorological data, this study evaluated MSWX precipitation, temperatures, and calculated potential evapotranspiration (PET) by direct comparison with observed data. The hydrological utility was performed by the TUW model based on four scenarios blending observed and MSWX as meteorological forcing. Observed meteorological data were provided from 23 stations and observed streamflow was prepared at the outlet of Karasu basin. Moreover, three performance indicators were utilized for MSWX dataset analysis, and the evaluation was done based on five hydrological years (2015–2019). The following main conclusions can be drawn from this study:

- Overall, MSWX-based temperature data show high performance (median of KGE > 0.90) when compared with observed temperatures directly. Among temperatures, the MSWX average temperature shows the highest performance (median of KGE; 0.97) on the regional scale. Compared to other meteorological forcing, MSWX-based precipitation shows lower performance (median KGE of 0.53) for the daily time step at the regional level. However, this is the only dataset which has the highest performance compared to previous studies over the Karasu basin for the daily time step. In the same way, MSWX based calculated PET shows high performance (median KGE of 0.93) for the study area.

- MSWX precipitation shows high detectability strength for moderate (5–20 mm/day) precipitation and its detectability strength decreases for heavy (20–40 mm/day) and violent (>40 mm/day) precipitation. MSWX precipitation showed a higher frequency of occurrence for light (1–5 mm/day) precipitation compared to observed precipitation, and the high frequency of occurrence directly affected MSWX detectability strength for the mentioned precipitation threshold.

- Considering 95PPU in model parameters, when the model is calibrated entirely by observed data (Scenario 1), it shows a relatively smaller range of uncertainty (95PPU) for most model parameters, whereas Scenario 4, which is entirely based on MSWX dataset, shows a slightly wider uncertainty bound (95PPU) for some parameters comparatively.

- When observed precipitation is considered for model calibration (Scenario 1, 3), the model shows high performance for streamflow simulation, where Scenario 2 and Scenario 4 show lower streamflow reproducibility, especially for the validation period. This can be attributed to the bias in MSWX precipitation datasets (Section 3.2), which shows direct effects on streamflow simulation. However, considering MSWX precipitation in different scenarios, it shows acceptable performance for streamflow reproducibility.

This study confirms the outperformance of the MSWX dataset compared to previous studies that dealt with climate data validation over the Karasu basin [24,25,40]. The novelty of this study was raised for the basins which were partially or fully ungauged. Hence, we considered four possible scenarios based on the basins' data scarcity. Furthermore, there are some precipitation datasets or PET individually, which can be an alternative for data scarce basins, and we recommend the combination of meteorological forcing from different available resources for streamflow simulation. However, our findings provide a valuable contribution to the existing literature for regions with complex topography and data scarcity, such as mountainous areas of Turkey and other similar regions of the world.

**Author Contributions:** H.H. contributed to the methodology, data analyses, and drafted the first manuscript. A.A.S. helped in conceptualization, supervision, and editing. All authors have read and agreed to the published version of the manuscript.

**Funding:** This study was partly supported by the Eskisehir Technical University Scientific Research Project (Project No: 20DRP214).

**Institutional Review Board Statement:** Not applicable.

**Informed Consent Statement:** Not applicable.

**Data Availability Statement:** Publicly available datasets were analyzed in this study.

**Acknowledgments:** The authors would like to acknowledge the State Meteorological Directorate (MGM), the State Hydraulics Works (DSI) of Turkey, and all other organizations providing data for this study. We also appreciate the valuable comments and suggestions of the Anonymous Reviewers.

**Conflicts of Interest:** The authors declare no conflict of interest.

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
