# Peer review of "Integrating Meteorological Forcing from Ground Observations and MSWX Dataset for Streamflow Prediction under Multiple Parameterization Scenarios"

_water, doi:10.3390/w14172721_

Round 1
Reviewer 1 Report
A good article with appropriate innovation that can be used as a reference method for flow modeling in other watersheds. Its publication is recommended.
Author Response
The authors would like to thank the reviewer for taking valuable time and effort to review the manuscript and appreciate the interest for the topic and developed methodology.
Reviewer 2 Report
This paper compares the accuracy of the MSWX dataset with the field observation dataset for precipitation, temperature, and potential evapotranspiration and clarifies the accuracy of the MSWX dataset. The paper also clarifies the reproducibility of river flow by conducting hydrological analysis using four scenarios based on field observation and MSWX datasets. There is nothing new in the methodology. The results obtained from the analysis are as expected, with the method using only field observation data sets being the most accurate. The study also shows the performance of the MSWX dataset for a specific watershed and does not reveal anything about the general performance of the MSWX dataset. While it is commendable that the study provides valuable information for selecting the most accurate combination of data depending on the availability of data, the paper only shows the performance of the MSWX dataset under a limited set of conditions. The reviewer is concerned that the paper only demonstrates the performance of the MSWX dataset under limited conditions, which is a critical problem of the paper. In order to resolve this issue, it would be most effective to conduct a similar analysis in more watersheds and examine the performance of the MSWX dataset in other watersheds with different geology, topography, and climate. If this is not possible, comparisons with other studies may also be effective. Since the introductory section contains many literature reviews, the authors could obtain useful information by comparing the results with those obtained using other datasets or by examining the performance of the MSWX dataset in other watersheds.
Some specific comments are as follows
Line 186.
What is HBV?
Line 223, equation (1), lines 225 to 227
The font of variables should be unified.
Lines 236 to 239
What are the units of M, F, H, and CN?
Lines 264 to 268
The reason for the high minimum temperature and low maximum temperature in the MSWX dataset is attributed to the weather conditions in the basin and the open and flat topography, but the causal relationship is unclear. A detailed explanation is needed.
Lines 305-315
The author states that MSWX is capable of detecting daily rainfall of less than 1 mm, but less capable of detecting daily rainfall of greater intensity, but later states that its ability to detect moderate rainfall is greater than that of weak rainfall. These may be contradictory.
Lines 342-345, Figure 4
“the wider interquartile range in each box represents higher variation of a certain parameter during the optimization”
What is the physical implication of this sentence? The authors should add more explanation for the physical significance of the figure.
Line 365
“Scenario 1 and Scenario 3 show a narrower uncertainty bound”
What is the physical meaning of this sentence? What does the “narrow uncertainty bound” mean? The authors should add more explanation for the physical significance of the figure.
Figures 2c and 6
According to Figure 2c, the agreement between the MSWX dataset and the observation is rather poor. However, according to Figure 6, the agreement between streamflow simulated by the use of the MSWX dataset and that by the use of the observed dataset is not so bad. The authors should clearly explain why this is the case
Author Response
The authors appreciate the valuable comments/suggestions by the reviewer which are taken into detailed consideration both in response to reviewers and in the revised manuscript.

Reviewer 3 Report
Dear authors,
It is a great opportunity for the reviewer to review the very relevant paper by Hafizi and Sorman (2022). The reviewer feels that this paper brings a new methodology to evaluate the hydrological attributes' different sources and their integration or coupling. However, the authors should deal with minor revisions in the attached report before publication.
Good luck with your improved version.
Best regards,
Anonymous reviewer.

Author Response

(The authors gave the same response as above.)

Round 2
Reviewer 2 Report
The reviewer confirmed that the authors have adequately responded to the reviewer’s comments. The reviewer believes that the manuscript is acceptable in its present form.